# Long-Term Use of Proton Pump Inhibitors in Cancer Patients: An Opinion Paper

**DOI:** 10.3390/cancers14051156

**Published:** 2022-02-24

**Authors:** Jean-Luc Raoul, Julien Edeline, Victor Simmet, Camille Moreau-Bachelard, Marine Gilabert, Jean-Sébastien Frénel

**Affiliations:** 1Department of Medical Oncology, Institut de Cancérologie de l’Ouest, 44805 Saint-Herblain, France; camille.moreau-bachelard@ico.unicancer.fr (C.M.-B.); jean-sebastien.frenel@ico.unicancer.fr (J.-S.F.); 2Department of Medical Oncology, Centre E Marquis, 35000 Rennes, France; j.edeline@rennes.unicancer.fr; 3Department of Medical Oncology, Institut de Cancérologie de l’Ouest, 49055 Angers, France; victor.simmet@ico.unicancer.fr; 4Department of Medical Oncology, Centre Hospitalier de Cholet, 49300 Cholet, France; 5Department of Medical Oncology, Centre Hospitalier Universitaire Vaudois, 1011 Lausanne, Switzerland; marine.gilabert@chuv.ch

**Keywords:** proton pump inhibitors, cancer, tyrosine kinase inhibitors checkpoint inhibitors, drug interactions, efficacy

## Abstract

**Simple Summary:**

Proton pump inhibitors are frequently used in cancer patients to alleviate some symptoms, epigastric pain or heartburn. However, acid suppression decreases the absorption of some oral-targeted anticancer treatments (tyrosine kinase inhibitors, CDK4/6 inhibitors) and induces changes in the gut microbiome. Recent data are showing that these interactions have important clinical impacts and medical oncologists and patients must be aware of these possible interactions.

**Abstract:**

Multikinase inhibitors (MKIs), and particularly tyrosine kinase inhibitors (TKIs) and immune checkpoint inhibitors (CPIs), are currently some of the major breakthroughs in cancer treatment. Proton pump inhibitors (PPIs) revolutionised the treatment of acid-related diseases, but are frequently overused for epigastric pain or heartburn. However, long-term acid suppression from using PPIs may lead to safety concerns, and could have a greater impact in cancer patients undergoing therapy, like bone fractures, renal toxicities, enteric infections, and micronutrient deficiencies (iron and magnesium). Moreover, acid suppression may also affect the pharmacokinetics of drugs (at least during acid suppression) and decrease the absorption of many molecularly-targeted anticancer therapies, which are mostly weak bases with pH-dependent absorption. This type of drug-drug interaction may have detrimental effects on efficacy, with major clinical impacts described for some orally administrated targeted therapies (erlotinib, gefitinib, pazopanib, palbociclib), and conflicting results with many others, including capecitabine. Furthermore, the long-term use of PPIs results in severe alterations to the gut microbiome and recent retrospective analyses have shown that the benefit of using CPIs was suppressed in patients treated with PPIs. These very expensive drugs are of great importance because of their efficacy. As the use of PPIs is not essential, we must apply the precautionary principle. All these data should encourage medical oncologists to refrain from prescribing PPIs, explaining to patients the risks of interaction in order to prevent inappropriate prescription by another physician.

## 1. Introduction

Proton pump inhibitors (PPIs) are one of the most frequently prescribed drugs in the world, and are ranked in the top 10 of US national health-related drug expenditures [1]. These highly efficient drugs in “acid related diseases” are widely available, including “over-the-counter” and at low cost, and are frequently prescribed inappropriately outside of their proven indications (gastric and duodenal ulcer, reflux oesophagitis, prevention of gastrointestinal bleeding when combined with non-steroidal anti-inflammatory drugs, Zollinger–Ellison syndrome) and in long-term use. This overuse is estimated between 40% to 80% in different countries [2,3]. Fortunately, they are very well tolerated, but the initial phase of omeprazole development was stopped when it was shown that carcinoids (ECLome) developed in the oxyntic mucosa in rodents [4]. Nevertheless, in the last decade, growing concerns have emerged regarding their safety, with a large number of studies reporting long-term toxicity, including cancer (of gastric, pancreatic, liver and biliary tract location) [5]. Cancer patients are fragile and many receive long-term PPIs. In a prospective study in four French Comprehensive Cancer Centres, we show that more than a quarter of cancer patients used PPIs, mostly on a daily basis and in the long term [6].

Certain side effects of long-term PPI use may be of greater impact in cancer patients than in the general population. On the other hand, long-term suppression of gastric acidity can decrease the absorption, and thus the efficacy, of certain major oral anticancer drugs, as well as changing the composition of the gut microbiome, which also has an impact on the response to immunotherapy [7]. This means that a symptomatic treatment that is not mandatory but is easily removable, might not only produce side effects, but also worsens patients’ prognosis [8,9]. The use of PPIs in cancer patients is thus a real issue [10]. 

In this review, we aim to update these potential interactions between long-term use of PPIs and cancer patients and their treatment, as well as to propose some possible solutions for cancer patients suffering from heartburn. 

## 2. Systemic Toxicity Linked to Long-Term PPI Use with a Possible Impact in Cancer Patients

### 2.1. Dementia

This question is of particular importance as the use of PPI therapy peaks in older people and cancer predominantly affects the elderly. The biological rationale is based on vitamin B12 deficiency, interaction with certain brain enzymes, and enhanced brain beta-amyloid levels (decreased degradation by lysosomes) [1,11]. A large German prospective cohort study, using observational data, followed more than 73,000 participants over the age of 75 years and free of dementia at baseline. Patients regularly using PPIs (n = 2950) had a significant risk of incident dementia compared with those not using PPIs (HR = 1.44; 95%CI: 1.36–1.52) [12]. Four retrospective and prospective cohorts, however, did not confirm this association [5], which is considered weak [1] when using the Hill criteria (association or causation) [13].

### 2.2. Bone Fractures

It is now widely accepted that PPI use is a risk factor for the development of osteoporosis and osteoporotic fractures [14,15,16]. This can be due to malabsorption of calcium, secondary hyperparathyroidism, and vitamin B12 deficiency. This side effect can be of major importance in the cancer patient population which has accelerated bone loss because of their cancer management [17]. 

### 2.3. Renal Toxicities

In a population-based cohort, PPI use was associated with a 20–50% higher risk of incident chronic kidney disease, as well as of acute kidney injury [18]. Recently, it has been shown that in patients with chronic kidney disease, chronic use of PPIs accelerates progression of the kidney disease and increases mortality [19]. Another recent retrospective observational study confirmed that in adults with chronic kidney disease, the use of PPIs was associated with an increased risk of hospitalization and mortality [20]. Taking into account the nephrotoxicity (acute but also chronic) of many anticancer drugs, associating them with PPIs should certainly be avoided.

### 2.4. Enteric Infections

The use of PPIs reduces gastric acidity, leading to changes in the gut microbiome, in the same way that antibiotics do. It also decreases colonization resistance to esophageal candidosis and enteric infections including *Clostridium difficile*, *Campylobacter* and *Salmonella* [7]. On a population level, the effect of PPIs on the gut microbiome is more prominent than the effects of antibiotics [21]. These PPI-induced changes in the microbiome may have a clinical impact, particularly in terms of the development of *Clostridium difficile* infections in the general population. The use of PPIs is also associated with an increased risk of community-acquired pneumonia [22]. In the frail population of cancer patients, long-term PPI prescription may lead to a high risk of enteric infections.

### 2.5. Micronutrient Deficiencies

Gastrointestinal acidity is important for the absorption of minerals (iron, calcium, magnesium) and vitamin B12. Patients with gastrinoma needing long-term use of high doses of PPIs are a natural model for studying their long-term effects in humans [23]. In this population, long-term use of PPIs was not associated with a decrease in total body stores or iron deficiency [24]. However, in a randomised controlled study in patients with hereditary haemochromatosis, long-term administration of PPIs significantly reduced the volume of blood needed to be removed annually to maintain serum ferritin at 50 µg/L, and 7 days of PPIs significantly decreased absorption of non-haem iron from meat [25,26]. Nevertheless, the development of iron deficit anaemia in patients on long-term PPIs seems infrequent and it is always necessary to exclude other causes. Anaemia in cancer patients often has multiple causes; however, avoiding unnecessary PPIs could be a good policy. 

Hypomagnesaemia (decreased absorption and increased renal leaks) due to PPIs has been well documented and many dramatic cases have been reported [27]. In 2011, the US FDA released a warning about low serum magnesium levels associated with long-term PPI use. A cross-sectional study in hospitalised patients in Buenos Aires demonstrated that 36% of patients with chronic PPI use had hypomagnesaemia on admission [28]. Association with other drugs used in oncology, and sometimes themselves, the cause of severe hypomagnesaemia, such as cisplatin and EGF receptor antagonists (monoclonal antibodies and tyrosine kinase inhibitors), requires regular follow-up of magnesaemia.

## 3. PPIs and Oncologic Treatment Efficacy

The concomitant use of oral antineoplastic agents in patients who are long-term PPI users is a real concern because of the consequences of severe chronic acid suppression, as well as the modifications to the intestinal microbiome. 

Many papers have addressed the question of the effects of acid suppressive compounds (PPIs and H2 antagonists) on the bioavailability of oral anticancer agents. As TKIs are weakly basic, when the gastric pH is elevated (through the use of PPIs or H2 antagonists) the solubility and bioavailability of these drugs may decrease significantly [29,30]. This decreased bioavailability can sometimes be significant and associated with decreased efficacy. One review reported a major decrease in the oral absorption of crizotinib, dasatinib, erlotinib, gefitinib, lapatinib and pazopanib, and recommended avoiding concomitant use of PPIs or H2 antagonists [31]. A recent systematic review and meta-analysis of the use of gastric-acid suppressants and oral anticancer treatments supports the evidence of a possible negative impact of such combinations on survival outcomes [32].

In parallel, there is increasing evidence suggesting that the gut microbiome can modulate the host’s antitumor response and the response to immune checkpoint inhibitors. It has been shown that antibiotics can inhibit the clinical benefits of immune checkpoint inhibitors by modifying the composition of the gut microbiome [33]. PPIs decrease bacterial richness and induce changes in the gut microbiome; these effects are more prominent than the effects of antibiotics [21].

### 3.1. PPIs and Tyrosine Kinase Inhibitors (TKIs) 

TKIs are currently a major weapon in the anticancer arsenal. Oral administration, which is convenient for both patients and physicians, and major efficacy in many forms of cancer, explain why these new drugs are currently one of the major options in the fight against cancer. Most medical oncologists are aware of drug–drug interactions with PPIs, (Table 1) but PPIs are frequently prescribed by the primary care physician, and can even be purchased over the counter, resulting in “unknown” drug–drug interactions that can lead to a decrease in efficacy [34,35].

Gefitinib and erlotinib, both selective TKIs targeting the epithelial growth factor receptor, showed reduced absorption in cases of concomitant use with PPIs, [36,37] translating into a significant decrease in efficacy (overall survival and progression-free survival) in retrospective analyses [38,39]. In a large retrospective study of the concomitant use of TKIs and PPIs, nearly 1 in 4 older adults with cancer who received TKIs also received PPIs concomitantly, and this was associated with an increased risk of death—an increase of 21% in lung cancer patients receiving erlotinib and not associated with discontinued use of TKIs [40]. In this study, no impact was observed in the case of co-prescription of PPIs with sunitinib or imatinib, confirming previous results [41]. However, in a real world study, results on the use of PPIs and the impact on first-line sunitinib treatment outcomes are conflicting [41,42]. No impact on serum concentration with PPI use was demonstrated with osimertinib [43].

In a retrospective analysis of two prospective trials of pazopanib (one single-arm phase 2, EORTC 62043, and one placebo-controlled phase 3 study, EORTC 62072) in soft-tissue sarcoma patients, of the 333 patients receiving pazopanib, 59 received concomitant PPIs or antiH2; progression-free survival and overall survival were shorter in pazopanib patients receiving gastric antisecretory drugs (respectively 2.8 vs. 4.6 months and 8.0 vs. 12.6 months); these effects were not observed in the placebo group of patients [44]. 

Clinical pharmacology studies consider that exposure to lenvatinib, vandetanib, cabozantinib, alectinib, and regorafenib is not significantly modified by PPIs [35,45].

In hepatocellular carcinoma patients, studies have produced contradictory results; a nationwide cohort study from Taiwan compared patients who took TKIs (sorafenib, regorafenib, lenvatinib and cabozantinib) and were PPI users (n = 2196) with those who were not PPI users (n = 8013). The one-year cumulative incidence of overall mortality was significantly higher in the PPI users (71.3% vs. 61.8%; *p* < 0.001) and this was confirmed in multiparametric analysis [48]. Similar results were found in a single-centre experience in the UK [49]. However, in secondary analysis of a phase 3 study comparing sorafenib with sunitinib, of the 542 patients receiving sorafenib, 122 were also treated with PPIs at baseline. On univariate and adjusted analyses, no significant association between PPI use and either OS or PFS was identified [50].

### 3.2. PPIs and Other Anticancer Treatments

No known interaction was demonstrated between mTOR inhibitors, PARP inhibitors [51], and PPIs; data regarding BRAF/MEK inhibitors and larotrectinib were scarce but seemed negative [35].

The solubility of palbociclib was reduced at pH above 4 and coadministration with PPIs decreased both AUC and Cmax [52]. In metastatic breast cancer patients treated with palbociclib, the concomitant use of PPIs may have a detrimental effect on progression-free survival [53]. On the contrary, gastric pH did not influence the pharmacokinetics of ribociclib.

No pharmacokinetic interaction between PPIs and oestrogen receptor inhibitors has been described, but enzalutamide, an androgen receptor inhibitor, can decrease the PPIs’ plasma levels [54]. 

### 3.3. PPIs and Immunotherapy 

Recent works on preclinical models, confirmed in retrospective analyses, suggest that patients who received antibiotics around the time of the initiation of immune checkpoint inhibitors (ICI) experienced reduced clinical benefits [33,55]. However, in humans, the effects of PPIs are more prominent than the effects of antibiotics on the gut microbiome [21]. Numerous studies have thus addressed the problem of ICI efficacy in PPI users.

In a cohort of 112 melanoma patients treated with anti PD-1, significant differences were observed in the microbiomes of responders versus non-responders [56]. In a retrospective analysis from CheckMate 069, the objective response rate (and PFS) after immunotherapy (ipilimumab alone or combined with nivolumab) in patients receiving PPIs was half that of those not on PPIs [57]. 

In 2020, retrospective analysis using pooled data from one phase 2 and one phase 3 trial comparing atezolizumab (n = 757) with docetaxel (n = 755) in previously-treated non-small-cell- lung cancer (the POPLAR and OAK trials) showed that PPI use was associated with shorter OS and PFS in the atezolizumab population and not in the docetaxel population [58]. Individual participant data from two urothelial cancer trials (IMvigor210 and 211) testing the efficacy of atezolizumab were analysed retrospectively with regard to the concomitant use of PPIs (approximately 30% of patients). In the pooled group of patients receiving atezolizumab (n = 847), PPI use was a negative prognostic marker (for overall survival, progression-free survival and response rate); in the randomised trial, atezolizumab showed significant efficacy on OS versus chemotherapy (HR: 0.69; 95% CI: 0.56–0.84) for PPI non-users and no OS benefit (HR: 1.04; 95% CI: 0.81–1.34) for PPI users; the same results were observed for PFS and ORR [59]. The phase 3 trial, IMpower 150, compared in non-small cell lung cancers, three chemotherapy regimens, two composed of atezolizumab. In post hoc analysis (1202 participants, 441 receiving PPIs), PPIs use was independently associated with worse overall survival in the pooled atezolizumab arms (n = 748), but not in the third arm without ICI [60]. The OS effect of atezolizumab was negative for PPIs users (HR: 1.03; 95% CI: 0.77–1.36), while it was clearly positive for non-users (HR: 0.68; 95% CI: 0.54–0.86). The concomitant use of PPIs thus transforms a major breakthrough drug into a treatment that is inefficient. (Table 2).

In a recent Korean cohort study of 2963 patients treated with ICIs as the second line, for non-small cell lung cancer, 936 were concomitant PPIs users. After propensity score matching (1:1 ratio), 1646 were analysed. The use of PPIs was associated with a higher risk of mortality compared to non-use (HR: 1.28; 95% CI: 1.13–1.46) [61].

An Italian series evaluated the prognostic impact of concomitant treatments (antibiotics, PPIs, or corticosteroids), quantified by a drug score, in a large series of patients receiving pembrolizumab or chemotherapy for non-small cell lung cancer. This drug score had a predictive value for response rate, OS and PFS, essentially in the pembrolizumab cohort [62].

Recently, a meta-analysis of seven studies (3647 cancer patients) was reported. The authors concluded that PPIs’ use had a detrimental effect on the efficacy of ICI: PPIs’ use increased the risk of death by 39% and the risk of progression by 28% [63].

In Bordeaux University Hospital, between May 2015 and September 2017, 635 patients received CPI for cancer. The authors analysed the influence of comedications (including PPIs) on the anti-tumour effect and safety of these CPI. PPIs were prescribed in 38% of these patients; the median OS of patients receiving PPIs was 9 months versus 26.5 months in those not receiving PPIs (HR: 1.70, 95%CI: 1.40–2.08) [64].

### 3.4. PPIs and Chemotherapeutic Agents

High doses of parenteral methotrexate are used in some forms of cancer and require strict drug monitoring. In a series of 74 patients receiving high dose methotrexate, it was demonstrated that co-administration of PPIs was associated with delayed elimination of methotrexate, as well as renal and liver dysfunction [65]. The mechanism is uncertain, but PPIs should be used cautiously with a high dose of methotrexate. 

In 2017, secondary unplanned analysis of the TRIO-013 trial comparing capecitabine and oxaliplatin (CapOx) with or without lapatinib in ERB2-positive metastatic gastroesophageal cancer aimed to determine if orally administered capecitabine or lapatinib were hampered by concomitant prescription of PPIs [66]. Of the 545 randomised patients, 229 (42%) evenly distributed patients received PPIs. In the placebo arm (receiving CapOx only), patients treated with PPIs had worse efficacy results (PFS, disease control rate, and OS) than those not receiving PPIs. The same authors conducted retrospective analysis of patients with stage II–III colorectal cancer who received adjuvant CapOx or FOLFOX in Edmonton, Alberta. Between 2004 and 2013, 389 patients, 214 receiving CapeOx and 175 receiving FOLFOX, met their inclusion criteria; respectively, 50 (23.4%) and 49 (28%) had concomitant PPIs. Three-year recurrence-free survival was significantly lower in the CapeOx-treated PPIs recipients than the non-PPIs recipients. This was not demonstrated in the FOLFOX-treated PPI recipients, but the differences were minor [67]. More recently, secondary analysis of six randomised controlled trials in patients with advanced colorectal cancer treated with fluoropyrimidines was conducted using individual patient data. Of the 5594 patients included, 902 received PPIs at trial entry. PPIs’ use was significantly associated with worse overall survival (pooled HR, 1.20; 95% CI, 1.03–1.40; *p* = 0.02) and progression-free survival (overall pooled HR, 1.20; 965% CI: 1.05–1.37; *p* = 0.009); this was particularly obvious for patients under 5FU and not among those receiving capecitabine; nor was it obvious for patients treated with other gastric antisecretory drugs (such as H2 antagonists). The authors concluded that clinicians should cautiously consider the concomitant use of PPIs in such patients. The mechanistic basis was unclear: impact on several transporters, modifications to intracellular pH, or something else [68]. Future studies are thus warranted as a series are accumulating on such possible interactions [69].

## 4. Conclusions

To conclude, the effect of PPIs on the efficacy of certain anticancer agents, particularly TKIs and CPIs, is a major issue in daily practice. In this opinion paper, we have put emphasis on articles showing the potential negative impact of such combinations and particularly on unplanned retrospective analysis from prospective studies, because we can expect that no randomized trial can be and will be conducted on this topic; moreover, PPIs are symptomatic treatments that can be replaced without any major risk of interactions. There are articles that did not find clinical interactions, particularly with CPIs [70,71,72], but we think that the precautionary principle must be applied until there is demonstration of the absence of clinical interaction. It is certainly of major importance that patients can be helped to stop taking PPIs after four weeks of treatment, except in cases of severe oesophagitis, previous bleeding, or Barrett’s oesophagus, [73] and ideally that prescriptions of PPIs be avoided for heartburn or epigastralgia. Some tricks, such as drinking acidic beverages (cola) with erlotinib could be proposed, but the best way is certainly to replace these long-lasting drugs with other therapeutic means [74]. If the use of acid-suppressive drugs is necessary, H2 antagonists (ranitidine) can be used and given 2 h after TKIs. Antacids can also be used 2 h before or after the drug [31]. The use of PPIs should be limited to TKIs with no proven interactions between absorption and intragastric pH. In patients treated with CPIs, the interaction is not due to drug absorption but rather to the alteration of the gut microbiome and we can suppose that the negative effect may also be observed after long-term use of H2 antagonists. In such cases, antacids are the best option, although on-demand use of PPIs or H2 antagonists may also be proposed.

## Figures and Tables

**Table 1 cancers-14-01156-t001:** Pharmacokinetic (PK) interactions between H2 antagonists (H2A) or proton pump inhibitors (PPI) and tyrosine kinase inhibitors; recommendations and demonstration of the clinical impact of such interactions.

Drug Name:	PK Interactions [31,46]	Recommendations [31,46]	Clinical Impact
Afatinib	NA	NA	NA
Alectinib	±	NA	NA
Axitinib	+	H2A: OK, PPI: OK	
Cabozantinib	±	H2A: OK, PPI: OK	
Crizotinib	0	H2A: OK, PPI: OK	
Dasatinib	++	H2A: OK, PPI: no	NA
Erlotinib	++	H2A: OK, PPI: no	YES [38,39,40]
Gefitinib	+++	H2A: no, PPI: no	YES [38,39,40]
Imatinib	0	H2A: OK, PPI: OK	
Lapatinib	+	H2A: no, PPI: no	
Lenvatinib	0	H2A: OK, PPI: OK	
Nilotinib	+	H2A: OK, PPI: OK	
Osimertinib	0	H2A: OK, PPI: OK	
Pazopanib	++	H2A: OK, PPI: OK	YES [44]
Regorafenib	0 [47]	H2A: OK, PPI: OK	
Sorafenib	0	H2A: OK, PPI: OK	Conflicting results [48,49,50]
Sunitinib	+	H2A: OK, PPI: OK	Conflicting results [40,41,42]
Vandetanib	+	H2A: OK, PPI: OK	

Pharmacokinetic interactions: NA: no data available; 0: definitively no interactions; ±:
conflicting results; +: possible interactions; ++: clear interactions; +++: major interactions. Recommendations: NA: no data available, OK: concomitant use possible; no: concomitant use strongly discouraged. Clinical impact of concomitant use: NA: no data available; YES: clinical impact demonstrated in clinical series; Conflicting results: clinical series showing different results.

**Table 2 cancers-14-01156-t002:** Overall survival results of 2 randomised controlled studies comparing atezolizumab vs. systemic chemotherapy with proton pump inhibitor (PPI) users versus non-users. HR: hazard ratio of overall survival of atezolizumab versus chemotherapy.

Trial	Subgroup	n	HR (95%CI)
POPLAR	PPI users	494	0.92 (0.75–1.44)
n = 1512 [58]	PPI non-users	1018	0.73 (0.62–0.85)
IMvigor 211	PPI users	330	1.04 (0.81–1.34)
N = 931 [59]	PPI non-users	601	0.69 (0.56–0.84)

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
