# Peer review of "Long-Term Use of Proton Pump Inhibitors in Cancer Patients: An Opinion Paper"

_cancers, 2022, doi:10.3390/cancers14051156_

Round 1

Reviewer 1 Report

This paper provides and important update about the long-term use of proton pump inhibitors together with anticancer therapies. The manuscript is clearly presented and well written. The goal of the study is well defined and the conclusions are corresponding to the findings.

Minor comments

Line 62. Please specify what are the side effects associated to the long-term PPIs use with greater impact in cancer patients compared to the general population.

Tables 1 and 2. Clearifying notes should be reflected in a footnote to each table. 

3.4 Section. PPIs and chemotherapeutic agents.

Please consider findings of the following study:

Performance of capecitabine in novel combination therapies in colorectal cancer. Fahima Danesh Pouya, Yousef Rasmi, Irem Yalim Camci, Yusuf Tutar & Mohadeseh Nemati. Journal of Chemotherapy, 2021 DOI: 10.1080/1120009X.2021.1920247

3.2 Section. PPIs and other anticancer treatments.

Please consider the following study:

Pharmacokinetic effects of proton pump inhibitors on the novel PARP inhibitor fluzoparib: a single-arm, fixed-sequence trial in male healthy volunteers. Investigational New Drugs (2021) 39:796–802 doi: 10.1007/s10637-020-01034-w

Author Response

Answers to Reviewer #1

Comments and Suggestions for Authors

This paper provides an important update about the long-term use of proton pump inhibitors together with anticancer therapies. The manuscript is clearly presented and well written. The goal of the study is well defined and the conclusions are corresponding to the findings.

We would like to thank the reviewer for these positive comments that will help us to improve our manuscript.

Minor comments

Line 62. Please specify what are the side effects associated to the long-term PPIs use with greater impact in cancer patients compared to the general population.

Modification has been done in the Abstract :

However, long-term acid suppression from using PPIs may lead to safety concerns, some of greater impact in cancer patients undergoing therapy, like bone fractures, renal toxicities, enteric infections, and micronutrient deficiencies (iron and magnesium).

Tables 1 and 2. Clarifying notes should be reflected in a footnote to each table. 

We agree with the Reviewer and have modified our legend and have added foot notes :

Pharmacokinetic interactions: NA: no data available; 0: definitively no interactions; +: possible interactions; ++: clear interactions; +++: major interactions.

Recommendations: NA: no data available, OK: concomitant use possible; no: concomitant use strongly discouraged

Clinical impact of concomitant use: NA: no data available; YES: clinical impact demonstrated in clinical series; Conflicting results: clinical series showing different results.

3.4 Section. PPIs and chemotherapeutic agents.

Please consider findings of the following study:

Performance of capecitabine in novel combination therapies in colorectal cancer. Fahima Danesh Pouya, Yousef Rasmi, Irem Yalim Camci, Yusuf Tutar & Mohadeseh Nemati. Journal of Chemotherapy, 2021 DOI: 10.1080/1120009X.2021.1920247

We have added to References and added the following in lines 324-5: Future studies are thus warranted as series are accumulating on such possible interactions.69

3.2 Section. PPIs and other anticancer treatments.

Please consider the following study:

Pharmacokinetic effects of proton pump inhibitors on the novel PARP inhibitor fluzoparib: a single-arm, fixed-sequence trial in male healthy volunteers. Investigational New Drugs (2021) 39:796–802 doi: 10.1007/s10637-020-01034-w

We have added to References and added the following in lines 225-7: No known interaction was demonstrated between mTOR inhibitors, PARP inhibitors51, and PPIs; data regarding BRAF / MEK inhibitors and larotrectinib are scarce but seem negative.

Reviewer 2 Report

This manuscript entitled “Long-term use of proton pump inhibitors in cancer patients: an opinion paper” by Jean-Luc Raoul et al. provided a comprehensive review of PPI in cancer patients and the impact on anti-cancer treatment. They considered PPI significantly influence the efficacy of TKI or CPI in cancer patients.

  1. The major concerns of previous studies done in a retrospective nature even for some clinical trials. Negative findings (no association) were found in some studies. However, the authors only included the studies with positive findings. I considered this may result in misleading readers.

For examples:

The effect of proton pump inhibitor uses on outcomes for cancer patients treated with immune checkpoint inhibitors: a meta-analysis

Impact of Proton Pump Inhibitor Use on the Effectiveness of Immune Checkpoint Inhibitors in Advanced Cancer Patients

Concomitant Proton Pump Inhibitors and Outcome of Patients Treated with Nivolumab Alone or Plus Ipilimumab for Advanced Renal Cell Carcinoma

   For this reason, I consider the conclusion is assertive.

  1. In addition, the duration, dosage of PPI may influence the efficacy of anticancer drugs but the authors did not provide such evidence.

  1. In terms of endpoints, ORR and PFS are better than OS to reflect the impact of PPI in such treatment. As OS is determined by too many factors, I don’t agree that this is a good endpoint in some studies. Those taking PPI may have underlying comorbidities which may lead to unfavorable survival.

Minor:

  1. Is “eradication of 50 Helicobacter pylori” an indication for PPI? (Line 50)
  2. These impressive data were confirmed in 2021 in 2 other clinical settings. (Line 219) What are other 2 settings?
  3. We now have significant data showing that PPIs use dramatically influences the response to immunotherapy (Line 255) What is the data?

Author Response

ANSWERS TO REVIEWER #2

Comments and Suggestions for Authors

This manuscript entitled “Long-term use of proton pump inhibitors in cancer patients: an opinion paper” by Jean-Luc Raoul et al. provided a comprehensive review of PPI in cancer patients and the impact on anti-cancer treatment. They considered PPI significantly influence the efficacy of TKI or CPI in cancer patients.

  1. The major concerns of previous studies done in a retrospective nature even for some clinical trials. Negative findings (no association) were found in some studies. However, the authors only included the studies with positive findings. I considered this may result in misleading readers.

For examples:

The effect of proton pump inhibitor uses on outcomes for cancer patients treated with immune checkpoint inhibitors: a meta-analysis

Impact of Proton Pump Inhibitor Use on the Effectiveness of Immune Checkpoint Inhibitors in Advanced Cancer Patients

Concomitant Proton Pump Inhibitors and Outcome of Patients Treated with Nivolumab Alone or Plus Ipilimumab for Advanced Renal Cell Carcinoma

   For this reason, I consider the conclusion is assertive.

We fully agree with this comment but our goal was to write « an opinion paper » because we have now many warnings, most coming from retrospective and unplanned analysis from large scale randomized controlled trials. Some papers disagree but most came from unicentre small series including few (hundred) patients. But we agree with the reviewer, this was not clearly stated in our manuscript and we have added in the Conclusion :

In this opinion paper, we have put emphasis on articles showing the potential negative impact of such combinations and particularly on unplanned retrospective analysis from prospective studies, because we can expect that no randomized trial can be and will be conducted on this topic; moreover PPIs are symptomatic treatments that can easily be replaced without any major risk. There are articles that did not find clinical interactions, particularly with CPIs69-71, but we think that the precautionary principle must be applied until demonstration of the absence of clinical interaction.

We hope this will fit with Reviewer opinion.

  1. In addition, the duration, dosage of PPI may influence the efficacy of anticancer drugs but the authors did not provide such evidence.

 For sure, and this is crucial but in most series, « PPI use » was defined as use of PPI within a period of 30 days before and 30 days after treatment initiation. Retrospective nature of analysis precluded more precise definition. But in our prospective analysis (Raoul JL, et al. JAMA Network Open 2021) PPIs were used in 71.1% of the cases on a regular basis and in 67.2% at normal dosage (19% at double dose). But we only want to ring a bell…

  1. In terms of endpoints, ORR and PFS are better than OS to reflect the impact of PPI in such treatment. As OS is determined by too many factors, I don’t agree that this is a good endpoint in some studies. Those taking PPI may have underlying comorbidities which may lead to unfavorable survival.

 The reviewer is absolutely right, but we will use the same comment on our purpose : warning. Nevertheless, in most series ICIs/TKIs were used either in late line (IMvigor, pazopanib trials, …) or when used earlier, gave similar « positive » results on Progression Free Survival (IMpower, palbociclib series).

Minor:

  1. Is “eradication of 50 Helicobacter pylori” an indication for PPI? (Line 50)

It is in France, in association with antibiotics but I can understand that it may be different in other countries. This line is removed.

  1. These impressive data were confirmed in 2021 in 2 other clinical settings. (Line 219) What are other 2 settings?

IMvigor and IMpower trials but you are right, it was only to introduce the two other chapters and I have removed this sentence.

  1. We now have significant data showing that PPIs use dramatically influences the response to immunotherapy (Line 255) What is the data?

You are right, this conclusion sentence is not useful and is now removed.